

**Risk assessment of liquefaction-induced hazards using Bayesian**
**network based on standard penetration test data**
Tang Xiao-Wei[1], Qiu Jiang-Nan[2], Hu Ji-Lei [3,*]
[1]State Key Laboratory of Coastal and Offshore Engineering. Dalian University of Technology, Dalian,
116024, China. Institute of Geotechnical Engineering, Dalian University of Technology, Dalian,
116024, China
[2]Faculty of Management and Economics, Dalian University of Technology, Dalian, 116024, China
[3]School of Civil Engineering and Mechanics, Huazhong University of Science and Technology, Wuhan,
430074, China
Correspondence to: Hu Ji-Lei (hujl@hust.edu.cn)
**Abstract.** Liquefaction-induced hazards are responsible for considerable damages to engineering
structures during major earthquakes. Presently, there is not any effective empirical approach that can
assess different liquefaction-induced hazards in one model, such as sand boils, ground cracks,
settlement, and lateral spreading, due to the uncertainties and complexity of multiple related factors of
seismic liquefaction and liquefaction-induced hazards. This study used Bayesian network method to
integrate multiple important factors of seismic liquefaction, sand boils, ground cracks, settlement and
lateral spreading into a model based on standard penetration test historical data, so that the constructed
Bayesian network model can assess the four different liquefaction-induced hazards together for free
fields. In the study case, compared with the artificial neural network technology and the Ishihara and
Yoshimine simplified method, the Bayesian network method performed a better classification ability,
because its prediction probabilities of *Accuracy*, *Brier score*, *Recall*, *Precision*, and area under the curve
of receiver operating characteristic (*AUC* of *ROC*) are better, which illustrated that the Bayesian
network method is a good alternative tool for risk assessment of liquefaction-induced hazards.





Furthermore, the performances of the application of the BN model in estimating liquefaction-induced
hazards in the Japan's Northeast Pacific Offshore Earthquake also prove the correctness and reliability
of it compared with the liquefaction potential index approach. Except for assessing the severity of
hazards induced by soil liquefaction, the proposed Bayesian network model can also predict whether the
soil is liquefied or not after an earthquake, and it can deduce the process of a chain reaction of the
liquefaction-induced hazards and do backward reasoning, the assessment results from the proposed
model could provide informative guidelines for decision-makers to detect damage state of a field
induced by liquefaction.
**1    Introduction**
Prediction of liquefaction potential and assessment of liquefaction-induced hazards are two great and
closely related problems. The former problem just needs to answer whether the soil is liquefied or not
after an earthquake, whereas the latter problem not only needs to predict whether liquefaction-induced
hazards occur or not after soil liquefaction but also needs to assess the severity of different hazards
induced by liquefaction. Prediction of liquefaction potential of foundation soils is only the first step of
assessment of liquefaction hazards, which was well studied in the recent decades, such as the simplified
methods (Seed and Idriss 1971, 1982; Starks and Olsen 1995; Stokoe and Nazarian 1985) based on
standard penetration test (SPT), cone penetration test (CPT) and shear wave velocity measurement,
laboratory testing, numerical methods, and empirical liquefaction models (Goh 1994; Pal 2006; Toprak
et al. 1999) based on historical data. What is more important to the engineers is the effect of
liquefaction-induced hazards on foundation or superstructure after seismic liquefaction, but relatively
few studies have focused on it (Juang et al. 2005).
Field evidence of liquefaction-induced hazards in historical earthquakes mainly consisted of sand boils,
ground cracks, settlement and tilting of structures, and lateral spreading failures. Several methods have
been proposed to calculate these hazards, including numerical simulation method, laboratory test
approach, and field-testing-based method. Although recent advances in physical model experiment and
computational modeling of liquefaction-induced ground deformation were quite promising, challenges
remain yet on the critical unresolved problems, such as, without a perfect physical numerical model for
totally describing the complicated mechanic characteristics of soils, high expense and difficulty



Natural Hazards
and Earth System
Sciences

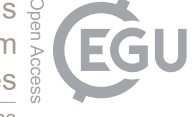

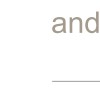

associated with obtaining and testing high quality undisturbed samples of loose sandy soils,
time-consuming, and large costs of manpower and material resources. Therefore, empirical liquefaction
models based on a historical database compiled from historical earthquakes are likely best suited to
provide a simple, reliable, and direct way to assess liquefaction-induced hazards for many years to come
in the field of geotechnical earthquake engineering (Zhang et al. 2002). As for the empirical liquefaction
method, liquefaction potential index (LPI) has been used to characterize liquefaction-induced hazards
worldwide that is proposed by Iwasaki et al. (1982). After that several approaches based on the LPI,
such as damage severity index (DSI) developed by Juang et al. (2005) that allowed for evaluation of the
severity of liquefaction-induced ground damage at or near foundations, the Ishihara inspired LPIISH
extended by Maurer et al. (2015) that was found to be consonant with observed surface effects and
showed improvement over LPI in mitigating false-positive predictions, and liquefaction severity
number (LSN) developed by Tonkin and Taylor (2013) that was developed following the liquefaction
damage observations from the Canterbury Earthquake Sequence to reflect the damaging effects of
shallow liquefaction on residential land and foundations. In addition, generalized analytical or empirical
techniques for estimating a single type of ground failures (e.g. settlement or lateral spreading) induced
by liquefaction have been proposed in the decades (Youd and Perkins 1987; Youd et al. 2002; Goh and
Zhang 2014; Ishihara and Yoshimine 1992; Zhang et al. 2002; Wu and Seed 2004; Cetin et al. 2009;
Juang et al. 2013). With the rapid development of computer technology and mathematics, many new
artificial intelligence methods for assessing liquefaction-induced ground deformation (Wang and
Rahman 1999; Baziar and Ghorbani 2005; Javadi et al. 2006; Garcia et al. 2008; Rezania et al. 2011)
have been developed based on the historical data. However, these methods cannot assess sand boils and
ground cracks, and they only can estimate a single type of hazards, e.g. lateral spreading or settlement.
Because there is not a generic model for calculating or assessing sand boils, ground cracks, lateral
spreading, and settlement simultaneously, and then evaluating the overall severity of hazards induced by
liquefaction after an earthquake at present, it is necessary to develop a generic model for assessing all
types of liquefaction-induced hazards at a given site subjected to an earthquake. The primary objective
of this paper is to use the Bayesian network (BN) technology to integrate soil liquefaction, liquefaction
potential index, the four types of hazards (ground cracks, sand boils, lateral spreading, and settlement)



induced by liquefaction and severity of liquefaction-induced hazards (describing the overall situation of
a site) into one model based on SPT historical data for deduce the process of chain reaction of the
hazards from an earthquake event to seismic liquefaction to liquefaction-induced hazards, that cover the
shortage of the existing simplified methods only assessing one single liquefaction-induced hazard. The
BN model is trained and tested separately using two different datasets of field case histories. The results
of the BN model for evaluation of liquefaction-induced hazards are compared with an artificial neural
network (ANN) model to verify its effectiveness and robustness using several performance indexes.
The remainder of this paper is organized as follows. At first, why used BN technology to assess the
hazards induced by seismic liquefaction was explained, and then construction of a BN model for
liquefaction-induced hazards was presented. Thirdly, the study case was used to verify the effectiveness
and robustness of the BN model compared with an ANN model and the Ishihara and Yoshimine
simplified method (Ishihara and Yoshimine, 1992) using several performance indexes, and the BN
model was applied for evaluating hazards induced by liquefaction in the 2011 Tohoku earthquake in
Japan that illustrated its availability. Finally, advantages and results of the BN model were discussed
with other methods, such as the ANN model and the LPI method. The conclusions and future works
were presented at the end of this paper.

## 2  Why Bayesian network?

Assessment of liquefaction-induced hazards is a complex engineering problem due to the heterogeneous
nature of soils, the participation of a large number of factors involved, and uncertainties of these factors.
The existed methods either were developed by statistics or only could assess a type of hazards such as
settlement or lateral spreading, and they hardly considered the effects of uncertainties on the
performances of the models, especially which purely relied on data-driven whereas ignoring the effects
of empirical knowledge or domain knowledge on the assessment of liquefaction-induced hazards.
However, the latest development in Bayesian network (BN) technology can combine empirical
knowledge and historical data that just right provides new opportunities to develop better performing
tools for complex problems in probabilistic terms such as the problem of liquefaction-induced hazards.
The BN is one of the most effective theoretical models for knowledge representation and reasoning
under the influence of uncertainty and high non-linear relationships among variables (Pearl 1988).
Firstly, BN can offer a rationalist and coherent theory for reasoning under the condition of various





uncertainties (e.g. uncertainties of parameters, models, and the domain knowledge) and complexities
that are described in terms of subjective beliefs or probabilities to reflect the interdependent relationship
between variables. Moreover, it can integrate different kinds of domain knowledge and multi-source
information or various quantitative factors and qualitative factors into a consistent system, and facilitate
multiple hazards and their interdependencies in a model. Especially, it allows not only sequential
inference (from causes to results) but also reverses inference (from results to causes) under conditions
of complete data, even incomplete data, and provide efficient frameworks for probabilistic updating and
the assessment of component performance when given new evidence.
In recent decades, the BN has been widely applied in risk analysis of the engineering field, such as
catastrophic risk (Li et al. 2010a; Li et al. 2010b; Li et al. 2012), earthquake risk damage (Bayraktarli et
al. 2005; Bayraktarli and Faber 2011; Bensi et al. 2009 and 2014), embankment dam risk (Zhang et al.
2011; Xu et al. 2011; Peng and Zhang 2012), landslide hazards (Song et al. 2012; Liang et al. 2012),
and soil liquefaction (Bayraktarli 2006; Hu et al. 2015), in which application of the BN in assessing
liquefaction-induced damages have never been involved before. An important sign is given by the fact
that, the numbers of relevant publications in this field from 2001 to 2015 were obtained by retrieving
with "BN" and "risk analysis" in Web of Science database annually increased from about 3 to 50 (as
shown in Fig. 1). Especially in recent five years, BN technology is generally popular with engineers and
researchers for assessment of risk analysis. The BN technology has been proven to be a robust method
for risk analysis.

## 3    A BN model for liquefaction-induced hazards

### 3.1 Probabilistic reasoning of BN

A Bayesian network combines the knowledge of graph theory and statistics theory that consists of
models and arcs or links with conditional probabilities. Its inference algorithms are based on the
Bayesian rule, the chain rule, and the conditional independence rule as follows
$$P(X \mid Y) = \frac{P(Y \mid X) \cdot P(X)}{P(Y)} ,$$    (1)
$$P(x_1, \cdots, x_n) = P(x_1) P(x_2 \mid x_1) \cdots P(x_n \mid x_1, x_2, \cdots, x_{n-1}) ,$$    (2)
$$P(x_1, \cdots, x_n) = \prod_{i=1}^{n} P(x_i \mid \pi(x_i)) ,$$    (3)



Where: P(Y) is called prior probability, P(X|Y) is one's belief for hypothesis X upon observing evidence
Y which is known as posterior probability, P(Y|X) is the likelihood that Y is observed if X is true.
$\pi(x_i)$ is a set of values for the parents of $X_i$.
A generic BN model for liquefaction-induced hazards (as shown in Fig. 2) is constructed with domain
knowledge to illustrate how to reason in the assessment of liquefaction-induced hazards. There are three
types of nodes in the BN model: (1) input nodes, they are soil parameters (SP), earthquake parameters
(EP) and field conditions (FC) that are factors of seismic liquefaction; (2) state nodes, they are
liquefaction potential (LP) and liquefaction potential index (LPI) that separately shows whether soil is
liquefied or not, and expresses how much the degree of soil liquefaction is; and (3) output nodes, they
are liquefaction-induced hazards (LH), such as lateral spreading, settlement, ground cracks, and sand
boils, which express the severity of liquefaction-induced hazards. The nodes are connected by twelve
arcs or links. In the risk assessment of liquefaction-induced hazards, if evidence comes from input
nodes, a posteriori probability or belief of the target variable (LH) being in a certain state (e.g. severe)
can be derived by the three above formulas of the BN method as follows:

$$P(\text{LH} = severe \mid SP, EP, FC) = \frac{P(\text{LH} = severe, SP, EP, FC)}{P(SP, EP, FC)}$$

$$= \frac{P(SP, EP, FC \mid \text{LH} = severe)P(\text{LH} = severe)}{P(SP, EP, FC)}$$

$$= \frac{\sum P(SP, EP, FC, LP, LPI \mid \text{LH} = severe)\sum P(\text{LH} = severe)}{\sum P(\text{LH}, LP, LPI, SP, EP, FC)}$$

$$P(\text{LH}, LP, LPI, SP, EP, FC) = P(SP) \cdot P(FC) \cdot P(EP \mid SP, FC) \cdot P(LP \mid SP, EP, FC)$$
$$\cdot P(LPI \mid \text{LH}, FC) \cdot P(\text{LH} \mid SP, EP, FC, LP, LPI)$$

**3.2 Construction of a BN model for liquefaction-induced hazards**
Strong earthquakes can cause liquefaction and therewith ground failures in the form of four types of
hazard, such as sand boils, ground cracks, settlement-induced tilting of structures, and lateral spreading.
Table 1 lists some factors of liquefaction potential (LP), liquefaction potential index (LPI), four types of
hazards induced by liquefaction and severity of liquefaction-induced hazards (SLH), in which LP and
LPI are used to describe the state of soil liquefaction, four types of liquefaction-induced hazards are
used to identify different damages and their severity after seismic liquefaction, and SLH is a
comprehensive index intergrading the four types of hazards indexes together that is used to describe the
overall severity of disasters after liquefaction. Also, Table 1 lists some empirical modeling methods that
can be used as domain knowledge to construct a BN model of liquefaction-induced hazards. Hu et al.





(2016) constructed a BN model for liquefaction potential (as shown in Fig. 3) that considered twelve
factors: the magnitude of the earthquake (ME), epicentral distance (ED), duration of the earthquake
(DE), peak ground acceleration (PGA), fines content (FC), soil type (ST), average particle size ($D_{50}$),
SPT number (SPTN), vertical effective stress ($\sigma_v'$), groundwater table (GT), depth of soil deposit (DSD)
and the thickness of the soil layer (TSL). In the seismic parameters, the bigger the magnitude of the
earthquake is and the nearer epicentral distance is, the longer duration of the earthquake is and the
bigger PGA is, the greater liquefaction potential is. In the soil parameters, anti-liquefaction behavior of
soil will largely change with different fines content, when the fines content increases within 30%, the
liquefaction strength decreases, but when the fines content is beyond 30%, the liquefaction strength
increases as fines content increases, and when the fines content is more than 50% (silt and sandy silt),
the soil is hardly liquefied. In addition, fines content can change the type of soil. Normally, purified clay
and silt cannot be liquefied, whereas poorly graded sand and silty sand are easily liquefied. The bigger
average particle size is, and the bigger SPT number is, the smaller the probability of soil liquefaction is.
In the field condition, the deeper depth of soil deposit is, the bigger vertical effective stress is, the
increase of pore water pressure is hard to overcome vertical effective stress, so soil liquefaction is not
easy to occur. In addition, shallow groundwater table and thin soil can partly reduce the probability of
soil liquefaction. So a state node (LPI) and output nodes (sand boils, ground cracks, lateral spreading,
settlement, and SLH) just need to be added into the exist BN model of liquefaction potential (as shown
in Fig. 3) based on the generic BN model in Fig. 2. A new BN model for liquefaction-induced hazards
(as shown in Fig. 4) was constructed according to domain knowledge of the hazards in Table 1. The
factor, ground slope that effects GC and LS, was not considered in the BN model of
liquefaction-induced hazards because the collected datum in this paper did not contain it.
Earthquake liquefaction-induced hazards are a chain reaction process originated from the earthquake
event to soil liquefaction to its pertinent hazards, in which different input values deduce different
liquefaction states and different the degrees of liquefaction, and then the outputs of the former system
(e.g. LP) are used as input information of the latter system to result in different hazard events (e.g. sand
boils, lateral spreading, et al.). The whole process of earthquake liquefaction-induced hazards can be
described as follow: at beginning of explosion earthquake, earthquake parameters, soil characteristics
and field conditions are considered as control variables, and their prior probabilities are calculated by
parameter learning. And then a posterior probability of the output variable (e.g. LP) can be reasoned for
estimating whether the event can be triggered or not. If the event occurs, its conditional probability is
replaced by the posterior probability that will be considered as evidence variable for input. Finally, a




posterior probability of the latter event (e.g. LP) will be calculated using the new conditional probability
of the former event for estimating its grade. Repeat the above process until grades of all hazards events
are identified at last.
**4   The study case**
**4.1 Dataset**
In this study, the data set including 442 SPT borings from post-earthquake in-situ tests at liquefied (245
SPT borings) and non-liquefied (197 SPT borings) sites in Taiwan, Japan, and the USA were collected.
332 SPT borings containing 184 liquefied sites and 148 non-liquefied sites were used to train the BN
model, and the rest 110 SPT borings were used to test the effectiveness and robustness of the BN model.
Because of incomplete data, e.g. the proportion of missing data for $D_{50}$ is about 15.2%, the proportion
of missing data for vertical effective stress is about 29.4%, and the proportion of missing data for the
thickness of soil layer is about 38.9%, expectation-maximization (EM) algorithm (Lauritzen 1995) was
used for training the 332 SPT borings data to obtain conditional probability table for the BN model due
to its more robustness compared with other algorithms and its suitableness for the data contained large
missing values. Briefly, the EM method is an iterative algorithm to find maximum likelihood estimation
or maximum a posterior estimation of parameters that repeatedly takes a Bayesian net and uses it to
obtain a better one by doing an expectation (E) step followed by a maximization (M) step until
convergence of the algorithm. In the E step, it uses regular Bayesian net inference with the existing
Bayesian net to compute the expected value of all the missing data, and then the M step finds the
maximum likelihood Bayesian net given the now extended data (e.g. original data plus expected values
of missing data). The data from the 1999 Chi-Chi earthquake in Taiwan ($M_w$=7.6) were downloaded
from                    http://www.ces.clemson.edu/chichi/TW-LIQ/In-situ-Test.htm                    and
http://peer.berkeley.edu/lifelines/research_projects/3A02/. Information of special "small magnitude"
was from the 1957 Daly City (California) earthquake ($M_w$=5. 3) of the USA and the 1987 Whittier
Narrows earthquake ($M_w$=5. 9) of the USA that refers to Cetin et al. (2000). The data from the 2011
Tohoku earthquake in Japan ($M_w$=9. 0) were provided by the research center for the management of
disaster and the environment, Tokushima University in Japan. Effects of liquefaction induced by these
earthquakes observed, including sand boils, settlement, ground cracks, and lateral spreading (as shown
in Fig. 5), resulted in destroying cropland, blocking the channels, and severe damages or collapses of
many buildings, highways, bridges, harbor facilities and other infrastructure components.
The grading standard of liquefaction and liquefaction-induced hazards according to domain knowledge





is as seen in Table 2, e.g. LPI is divided into four grades according to Iwasaki (1982), non-liquefaction
(LPI=0), slight liquefaction (0<LPI≤5), moderate liquefaction (5<LPI≤15), and serious liquefaction
(LPI>15). SLH is divided into four grades according to the experience of disaster in the field of
engineering that is described in detail as seen in Table 3. After that, according to the descriptions of
SLH, a statistic summary of liquefaction-induced hazards data is presented in Fig. 6. It can be seen that
(1) liquefaction doesn't have to induce hazards, but the occurrence of the liquefaction-induced hazards
is based on liquefaction, (2) LPI is not a good index for a description of the severity of
liquefaction-induced hazards because the efficacy of the LPI framework and accuracy of derivative
liquefaction hazards are uncertain, e.g. serious liquefaction of the LPI occurs in the none SLH in Fig. 6
(1) and slight liquefaction of the LPI occur in the severe SLH in Fig. 6 (4), but there is a rule that the
bigger the LPI is, the corresponding severity of the liquefaction-induced hazards is, (3) SB, S, GC, and
LS are macroscopical phenomena of hazards induced by liquefaction, and there is also a trend that the
bigger the values of these indexes are, the more severity of the SLH is, (4) the classifications for the
four different types of hazards in Fig. 6 almost accord with the descriptions of the field ground damage
status in Table 3.
Fig. 7 shows ratios of all influence factors for the severe status of the SLH. It can be easily seen that
most severe damage sites suffered from a big or super earthquakes with long loading, their epicentral
distance were near to the earthquake sources, and their PGA should be higher enough. As for soil
characteristics, pure sand or silty sand with moderate fines particles and moderate $D_{50}$ easily result in
severe damage, but few sites with gravel soil and sandy silt were also suffered from severe damage. The
damaging phenomena showed that even though gravel and sandy silt were not easy to be liquefied, if
they were liquefied in a big earthquake, they also resulted in severe damage. The small SPTN means
that the sand soil is so loose that it more easily results in settlement and lateral spreading after
liquefaction because loose sand is more easily compressed and flowing during seismic liquefaction. As
for field conditions, the shallowly deposited sandy soil layer is thin so that its effective stress is small,
and groundwater table is near to the ground surface, this kind of field is easily suffered from severe
damage. The above laws fit well with engineering practical experience. The sum of ratios of three
variables, such as $D_{50}$, $\sigma_v'$ and the thickness of soil layer, is not 1 due to data missing mentioned in
Section of Dataset.
**4.2 Performance indexes**
In this section, in order to comprehensively evaluate the performances of the two probabilistic models



for liquefaction-induced hazards, several performance indexes are used, which are *Accuracy*, *Prediction*,
*Recall*, area under the curve of the receiver operating characteristic (*AUC* of *ROC*), and *Brier score*. The
details of these indexes are briefly introduced as follows.
The *Accuracy* is a measure of the percentage of correctly classified instances for each class, which is a
widely used metric for measuring the overall performance of a classifier. For instance, the *Accuracy* is
equal to 0.9 that means 90% of data can be correctly classified. However, it does not mean that the
accuracies of each class are all 90%, perhaps the accuracy of one class is high but others are very low.
Therefore, the evaluation of the predictive capability based on the *Accuracy* alone can be misleading
when a class imbalance exists in a data set. Other indexes, *Precision*, *Recall*, and *AUC* of *ROC*, should
be used to further measure the performance of each class for a model or a classifier.
The *Recall* refers to the probability of detection of a class, and it measures the proportion of correctly
predicted positive instances among the actual positive ones. If a classifier can achieve a higher *Recall*
for a class that means it can detect more positive instances of the class. The *Precision* refers to the
proportion of true positives among the instances predicted as positive for a single class, but it cannot
measure how the classifier detects the actual positive instances. A classifier with high *Precision* but with
a lower *Recall* is less useful because it cannot detect significant positive instances, especially in the risk
assessment, security and warning are major concerns, a good classifier should detect more positive
instances and relatively high accuracy of prediction, namely, have a high Recall and acceptable
*Precision*.
The ROC curve is a graphical plot developed by the false positive rate (the proportion of all negatives
that still yield positive test outcomes) on the x-axis and the true positive rate or *Recall* on the y-axis that
can present an overly optimistic view of an algorithm's performance. The *AUC* of *ROC* is the area
between the horizontal axis and the *ROC* curve that is a comprehensive scalar value representing a
classifier's expected performance. The *AUC* of *ROC* is between 0.5 and 1, where a value close to 0.5 is
less precise, while a value close to 1.0 is more precise. Therefore, the bigger the AUC of *ROC* is, the
better prediction the performance of the classifier can get.
*Brier score* is used to measure the quality of probabilistic forecasts for discrete events, which was
proposed by Brier (1950). Suppose that on each of $n$ occasions an event can occur in only one of $r$
possible classes and on one such occasion, $i$, the forecast probabilities are $f_{i1}, f_{i2}, \cdots, f_{ir}$, that the event
will occur in classes $1, 2, 3, \cdots, r$, respectively. The *Brier score* ($B$) can be defined by



$$B = \frac{1}{n} \sum_{j=1}^{r} \sum_{i=1}^{n} \left( f_{ij} - E_{ij} \right)^2 ,$$  (4)
Where: $\sum_{i=1}^{r} f_{ij} = 1, i = 1,2,3,\cdots,n$ . $E_{ij}$ takes the value 1 or 0 according to whether the event occurred in
class $j$ or not. For instance, in the study case of this paper, 110 SPT borings used for testing ($n$=110),
SLH has four classes, none, minor, medium, and severe ($r$=4), in which a probability or confidence
statement ( $f_{ij}$ ) was made for each SPT boring instance. The *Brier score* is between 0 and 2, where the
minimum *Brier score* is 0 for perfect prediction and the maximum value is 2 for the worst possible
prediction.
**5 Results**
**5.1 Comparison of predictive results**
The comparisons of probability predictive results of the two models, the BN model (in Fig. 4) and the
ANN model used same parameters, are shown in Table 4. Comparing the probability of *Accuracy*,
except for LS, the *Accuracy* values of the BN model for the other types of hazards and SLH are
obviously more than the ANN's, and comparing the *Brier score*, its values of the BN model except for
LS and SLH are less than ANN's, which both illustrate that the overall performances of the BN model
are better than the ANN model. As for each type hazard induced by liquefaction and SLH, *Recall*,
*Precision*, and *AUC* of *ROC* of BN model for each class are almost more than the ANN's, which also
represents that BN model is better than the ANN model. Therefore, the proposed BN approach is better
than the ANN technology, and its performances are acceptable for monitoring and forecast of seismic
liquefaction-induced hazards. In addition, comprising the computation time, the BN model using the
EM algorithm outperforms the ANN model contained 20 hidden layers using radial basis function,
requiring 36 iteration steps (about 19.8 CPU seconds) to converge to a stable state, therefore the
convergence rate of the BN model is faster than the ANN model in this paper.
Furthermore there are not any effective simplified methods for estimating ground cracks and sand boils,
and simplified methods for calculating lateral spreading, e.g. Bartlett and Youd (1995), Wang and
Rahman (1999), Goh et al. (2014), et al. that need the free face ratio or the ground slope, however, these
two factors were not contained in collected datum in this paper. Therefore, ground cracks, sand boils
and lateral spreading cannot be estimated by simplified methods, while settlement can be calculated by
the I&Y simplified method proposed by Ishihara and Yoshimine (1992) as shown in Table 4. It can be





clearly seen that predictive results of data-driven methods, such as BN and ANN, are better than I&Y
simplified method, but the simplified method can get a certain value (as shown in Fig. 8) rather than an
interval value or a probability. In addition, the simplified method was only constructed by a relationship
among relative density, the factor of safety against liquefaction ($F_L$) and volumetric strain ($\varepsilon_v$) to
estimate the settlement of a site, in which the factor of safety against liquefaction was obtained by
integrating earthquake intensity and SPTN by empirical formulas or empirical coefficient so that it
might cause calculation error and result in considerable prediction error, such as the predictive result of
small settlement as shown in Table 4, the precision of the simplified method is only 0.069. However, the
data-driven methods can integrate multiple factors of liquefaction-induced hazards into a model so that
they can get better predictive performances than simplified methods.

## 5.2 Casual reasoning using the BN model

Based on the developed BN model, probabilities of the liquefaction-induced hazards were inference
through causal reasoning in this section. The third column in Table 5 shows the posterior probabilities of
all grades of LP, LPI, and its induced hazards. It can be seen that when the input information about
earthquake parameters, soil characteristics, and field conditions are not known, the probabilities of all
grades of in each output variable are no big different except for LS and SB due to its serious imbalance
data in different grades. However, when a site is determined to be liquefied and the probability of yes
status of LP is changed into 100%, it can be seen in the fourth column that probabilities of none status
of LPI and all hazards decrease in some extent but probabilities of other states in LPI and all hazards
increase largely. Furthermore, if the site is serious liquefied, the probability of yes status of LP and
serious status of LPI are changed into 100% as seen in the fifth column in Table 5, the probabilities of
grades (except none status) of all hazards continue increasing, in which GC occurs with 66.1%
probability, serious sand boil occurs with 69.6% probability, the probability of Big LS is 9.5%, the
probability of big settlement is 49.8%, and the probability of severe of SLH is 64.1%. It shows that
liquefaction-induced hazards are much more severe in the serious liquefied site. And then
macro-liquefaction phenomena, such as GC and serious SB, are observed, probabilities of the big status
of other hazards continue increasing slightly as seen in the sixth column in Table 5 so that the predictive
results are more close to the actual situation. Therefore, according to the above deduce process, the BN
model can calculate the posterior probability of LP based on conditional probabilities of input variables
for estimating whether a site is liquefied or not, if it is liquefied, its posterior probability will be
considered as input information for the latter variable's prediction. Reasoning step by step like that can
give all predictive results of liquefaction-induced hazards. In addition, when the prior probabilities of all



input variables, such earthquake parameters, soil characteristics and field conditions, can be determined actually at first, the predictive performances of all hazards will be improved to a large content. For instance, a site suffered from a super earthquake with long duration, survey shows that the SLH is severe with big settlement, no lateral spreading, serious sand boils and ground crack, the input variables of the site are that the epicentral distance is near, PGA is higher, soil type is sand with some fine particle, its $D_{50}$ is medium, SPT number shows the sand is loose, $\sigma_{v'}$ is small, groundwater table is shallow, both depth and thickness of the sand layer are moderate. The reasoning probability value of LP is 99.9%, LPI is identified as serious with 43.8% probability, GC doesn't occur with 51.4% probability that doesn't match with the survey result, SB is identified as many with 76.5% probability, LS is identified as none with 85.0% probability, settlement is identified as big with 53.1% probability, and SLH is identified as severe with 52.6% probability according to the input information. After that the site can be determined as a liquefied area with serious liquefaction degree, so LP is intended to be 100% probability, and the probability of serious state of LPI is also intended to be 100%, then the probabilities of all hazards are changed, GC occurs with probability 100% that matches with the survey result, LS is identified as none with 100% probability (increased by 15%), settlement is identified as big with 100% probability (increased by 46.9%), and SLH is identified as severe with 100% probability (increased by 47.4%).

**5.3 Diagnostic reasoning using the BN model**

In order to detect what kind of situations more easily result in severe damage of a site, the most probable explanations of LP (Yes), LPI (Serious), GC (Yes), SB (Many), LS (Big) and S (Big) are inference using the diagnostic reasoning capabilities of the BN model, the results are shown in Table 6. It can be seen that the loose silty sand (medium $D_{50}$) contained moderated fines particles deposits shallowly (small $\sigma_{v'}$) on a site with a low underground water level that is suffered from a super earthquake with moderate duration, and the epicentral distance is moderate, which is more easily subjected to be liquefied. The most probable explanations of GC and many SB are same as LP's under conditions of serious or moderate soil liquefaction, but the most probable explanations of big LS and S are tiny different from LP's, such as PGA and sandy soil, the reason is that big LS and big settlement need more seismic intensity than occurrences of sand boils and ground cracks, and sand is more easily flowing and compressed after liquefaction than the sand containing fines particles. In addition, occurrences of big LS and big settlement often accompany by many sand boils, while ground cracks may or may not occur. The above results are according with the analysis results in Fig. 7. In addition, If soil characteristics, field conditions, and hazards of the field are known, how strong the earthquake intensity (status of magnitude of the earthquake, duration of earthquake, PGA and epicentral distance)





resulting in the liquefaction-induced hazards of the field can be estimated using backward inference
ability of the BN method, that can provide some references for aseismatic design.
**5.4 Sensitivity analysis of liquefaction-induced hazards**
Sensitivity analysis can detect how much each factor has an impact on the target variable. In this section,
mutual information is used to assess sensibility that is a measure of the mutual dependence between the
two variables. The results of mutual information for different liquefaction-induced hazards are
computed separately in the BN model as shown in Table 7. It is shown that the thickness of soil layer is
the most sensitive variable for GC, and relative important factors are the depth of the soil layer, $D_{50}$, and
the duration of earthquake compared to the rest other factors. As for SB, groundwater table is the most
sensitive variable, and relative important factors are the thickness of the soil layer, SPTN, the duration
of the earthquake, PGA, depth of soil layer, and $\sigma_v'$ compared to the rest other factors. As for S, PGA is
the most sensitive variable, and relative important factors are SPTN, the duration of the earthquake, and
depth of soil layer compare to the rest other factors. As for LS, PGA is also the most sensitive variable,
and relative important factors are $D_{50}$, the thickness of the soil layer, depth of soil layer, and soil type
compared with the rest other factors. These results are highly consistent with the domain knowledge in
Table 1. When comprehensive comparisons with the most sensitive factors and relative important
factors of the four types of liquefaction-induced hazards and SLH are conducted, the duration of
earthquake, PGA, SPTN, depth of soil deposit, and the thickness of soil layer are more important than
other factors, because their overlap sections are more than three times. In these five factors, a
combination of SPTN and the earthquake intensity described by the duration of earthquake and PGA
can detect the degree of soil liquefaction. And then the depth of soil deposit and the thickness of soil
layer combine with relative density (determined by SPTN) based on the degree of soil liquefaction to
result in the soil volumetric strain. Consequently, liquefaction-induced hazards, e.g. settlement and
lateral spreading, can be estimated. Therefore, in order to mitigate seismic liquefaction-induced hazards
of a filed, dispose of relative density of sandy soil, the depth of sandy soil deposit and the thickness of
sandy soil layer are the crucial factors.
**6   Application of the BN model**
The above BN model is applied to assess the liquefaction-induced hazards in the Japan's Northeast
Pacific Offshore Earthquake on March 11, 2011. The research regions are Ibaraki prefecture, Chiba
prefecture, Saitama prefecture, Kanagawa prefecture and Tokyo city that contain 196 investigation sites.
The assessment results of the SLH are shown in Fig. 9, in which the blue circle means little to none



liquefaction-induced hazards, the green circle means minor liquefaction-induced hazards, the yellow circle means medium liquefaction-induced hazards, the orange circle means severe liquefaction-induced hazards and the red circle means prediction error. In the 196 real fields, the prediction accuracies of four types of liquefaction-induced hazards, respectively, are 99.50% for lateral spreading, 81.63% for sand boils, 80.61% for settlement, 89.8% for ground cracks, and 84.1% for SLH. In addition, the prediction accuracies of four different levels of SLH (Little to none, Minor, Medium and Severe) separately are 79.83%, 84.62%, 81.25% and 79.83%, which prove the validity of the BN model in general. However, the prediction accuracies of the LPI approach (Iwasaki et al. 1982) for the four different levels of SLH are 36.96%, 8.82%, 68%, and 42.22% separately, which are much worse than the prediction results of the BN model.

In this earthquake, the areas with greater losses and a larger number of liquefaction sites are in Ibaraki prefecture and Tokyo city that are near to the sea compared with other places, which contained 78 sites with different degrees of hazards, and in which disasters of about 50 sites are medium or severe. It can be known from Table 3 that free site with medium disasters or severe disasters is easy to cause sand boils and ground cracks phenomenon, lateral spreading and settlement, which is the result of foundation failure, further to cause building damage and bridge collapse. Therefore the BN model of assessing liquefaction-induced hazards not only accurately assess range values of lateral spreading and settlement, quantity of sand boils, and ground cracks phenomenon, but also accurately predict the severity of hazards induced by liquefaction for foundation on the free site, and then qualitatively assess disasters of a building or other structures according to the engineering experiences relationship between disasters of the foundation and losses of upper structural, which provide guidelines for prevention and mitigation engineering of natural disasters.

## 7   Discussion

This paper constructed a probability model for liquefaction-induced hazards using BN technology. As a means of probability inference, BN technology offers several specific advantages over other methods in the evaluation of catastrophic, which primarily can support a good platform for integrating different kinds of hazards and their interdependencies into a consistent system (Li et al. 2010b). However, existing empirical methods for estimating hazards induced by seismic liquefaction, such as the empirical formulas constructed by Youd et al. (1987, 2002) and the multiple linear regression (MLR) model constructed by Goh and Zhang (2014) for estimating lateral spreading, the different simplified procedures for estimating the settlement proposed by Ishihara and Yoshimine (1992), Zhang et al.





(2002), Wu and Seed (2004), and Juang et al. (2013), which only can assess a single type of ground
failures, and cannot estimate ground cracks and sand boils. Another method, the LPI approach, can
quantify liquefaction severity of a site by providing a unique value for the entire soil column instead of
several factors of safety per layer. However, calibration of LPI to observe liquefaction severity is
limited, and the efficacy of the LPI framework and accuracy of derivative liquefaction-induced hazards
are uncertain (Maurer et al. 2014). When the value of LPI is big (LPI>15), whereas there may be not
phenomenon of settlement or ground cracks, but when the value of LPI is small (LPI<5), whereas a
serious sand boils with a long duration and wide scale, lateral spreading and severe subsidence occur, so
the real SLH are largely not consistent with the prediction results of the LPI approach, which can be
found in Fig 6 and in the prediction results of the LPI approach in the section of Application of the BN
Model. In fact, LPI only can reflect the degree of liquefaction in a site, but cannot detect real situations
of ground damage, the relation between LPI and the types of liquefaction-induced hazards was not
being examined systematically that might be a qualitative relation in some extent.
Comparing the two methods, the BN method and the ANN method, although both of them are
supervised learning methods, the BN method is a generative model, whereas the ANN method is a
discriminative model, therefore, the BN method can obtain the joint probability distribution of the
parameters so that it can describe distributions of data in statistical terms and combine with a strong
probabilistic theory enabling it to give an objective interpretation, and its computation time is faster
than discriminative models such as the ANN method, even when sample size increases quickly, the BN
method can quickly converge to the true model, especially, when the data contains hidden parameters,
the BN method can still develop a robust model, but the ANN method cannot do it (Correa et al. 2009).
As for the models, in the BN model, each node means a random variable that has actual meaning, and
the arrow between two nodes means causation, but each node in the ANN model is not random variable
that has not actual meaning, and the arrow between two nodes just means weighting functional
relationship that has not other meanings, such as causation or logistic relationship, making it difficult to
explain the results. In addition, except predicting the different hazards induced by liquefaction, the
constructed BN model in this paper also can predict liquefaction potential, the *Accuracy* of liquefaction
potential is 0.80 using the testing data in the dataset. If using the ANN technology, a new model should
be reconstructed by studying the training data to predict the *Accuracy* of liquefaction potential, whereas
the BN model can predict directly without retraining. Especially, the BN method can reason forward
and reason backward, such as assessing hazards induced by liquefaction with given earthquake
parameters, soil parameters, and field conditions, or predicting what kind of soil properties and field

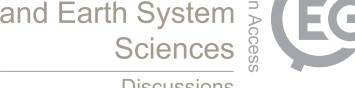



conditions are when hazards are known after an earthquake, whereas the ANN method just only can do
forward reasoning.

## 8    Conclusions and future work

Given uncertainty and complexity of liquefaction-induced hazards, this paper developed a generic BN
model for estimating risk assessment of different hazards induced by seismic liquefaction based on
historical disaster data that provides a platform integrating a variety of information sources from
different fields and facilitates different hazards induced by liquefaction into a model. The findings in
this paper are as follows:
(1) After comparing with ANN technology using several performance indexes, the BN model achieved
a better *Accuracy* and a better *Brier score* for overall performance, and a better *Recall*, a better
*Precision*, and a better *AUC* of *ROC* for each damage state (e.g. sand boils, settlement et al.) and its
computation time is faster than the ANN method. This illustrates that the BN method is suitable for risk
assessment of liquefaction-induced hazards influenced by complex multiple factors. Comparing with
the I&Y simplified method for estimating settlement, the data-driven methods (BN and ANN) are
superior to it. Furthermore, the performances of the application of the BN model in estimating
liquefaction-induced hazards in the Japan's Northeast Pacific Offshore Earthquake also prove the
correctness and reliability of it compared with the LPI approach.
(2) The BN model can deduce the process of a chain reaction of the liquefaction-induced hazards and do
backward reasoning, such as inference from input variables (earthquake parameters, soil characteristics
and field conditions) to soil liquefaction to different hazards events or from soil liquefaction to different
hazards events to input variables. In addition, the most probable explanations of LP, Serious LPI, GC,
many SB, Big LS and Big S in the BN model show that the loose silty sand or sandy soil (medium $D_{50}$)
contained moderated fines particles deposits shallowly (small $\sigma_v'$) on a site with low underground water
level that is suffered a super earthquake with moderate duration, and the epicentral distance is moderate,
it is more easily suffered to be liquefied and severe hazards induced by liquefaction.
(3) After sensitivity analysis for liquefaction-induced hazards separately, the most sensitive factors of
different hazards are different. When comprehensively comparing with these important factors of the
four types of liquefaction-induced hazards, the duration of the earthquake, PGA, SPTN, depth of soil
deposit, and the thickness of soil layer are more important than other factors, which contribute to the
soil volumetric strain.



Because the occurrence of liquefaction may cause no damage, little or severe damage to the ground
surface or infrastructure, the constructed BN model in this paper can accurately assess severity of
hazards after seismic liquefaction, and provides guidelines to engineers with which fields need to be
dealt with, rather than dealing with all fields with liquefaction, thus, large of costs will be saved. In this
study, more historical data will be collected to update the conditional probability table to improve the
BN model, especially collecting historical data contained statues of small and medium of lateral
spreading due to a lack of them in this paper's data, and then the nodes of utility and decision actions
will be trying to add into the BN model to extend it, so that the new model can test how different actions
will result in different hazards and different expected utilities of loss. The results can serve as significant
information for decision making in the earthquake resistance and hazard reduction.

## Acknowledgments

The work presented in this paper was part of research sponsored by the National Science Council of the
People's Republic of China under Grant No. 2011CB013605-2. The writers gratefully acknowledge Pro.
Uzuoka Ryosuke from the research center for the management of disaster and environment of
Tokushima University in Japan who provided the SPT data of the 2011 Tohoku earthquake in Japan for
this study.



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





1   Table 1. Factors of liquefaction and its induced hazards and empirical modeling

2   methods.

| Category | Liquefaction and its induced hazards | Factors | Empirical methods |
|---|---|---|---|
| Liquefaction state | Liquefaction potential (LP) | The magnitude of earthquake, epicentral distance, the duration of earthquake, peak ground acceleration (PGA), fines content, soil type, average particle size ($D_{50}$), SPT number (SPTN), vertical effective stress ($\sigma_v'$), groundwater table, depth of soil deposit, and the thickness of soil layer | Hu et al. (2016) |
| | Liquefaction potential index (LPI) | LP, depth of soil deposit, and the thickness of soil layer | Iwasaki et al. (1982); Maurer et al. (2015) |
| Liquefaction -induced hazards | Sand boils (SB) | LP, LPI, depth of soil deposit, the thickness of soil layer, and groundwater table | Bardet and Kapuskar (1993) |
| | Ground cracks (GC) | LP, LPI, D50, depth of soil deposit, the thickness of soil layer, and ground slope ($\theta$) | Youd (1984) |
| | Lateral spreading (LS) | LP, LPI, PGA, The magnitude of earthquake, epicentral distance, depth of soil deposit, the thickness of soil layer, D50, and $\theta$ | Bartlett and Youd (1995); Wang and Rahman (1999); Goh et al. (2014) |
| | Settlement (S) | LP, LPI, PGA, depth of soil deposit, the thickness of soil layer, soil type, LS, SB | Zhang, Robertson, and Rrachamn (2002); Cetin et al. (2009); Juang et al. (2013) |
| Comprehensive index | Severity of liquefaction-induced hazards (SLH) | LP, LPI, SB, GC, LS, S | - |





1    Table 2. Grading standard for liquefaction and liquefaction-induced hazards.

| Factor | No. of grade | Grade | Data number | Range |
|---|---|---|---|---|
| Liquefaction potential | 2 | None | 245 | - |
| | | Yes | 197 | - |
| Liquefaction potential index | 4 | Non-liquefaction | 145 | 0 |
| | | Slight liquefaction | 97 | $0<LPI\leq5$ |
| | | Moderate liquefaction | 106 | $5<LPI\leq15$ |
| | | Serious liquefaction | 94 | $15<LPI$ |
| Settlement (m) | 4 | None | 238 | 0 |
| | | Small | 23 | $0<S\leq0.1$ |
| | | Medium | 54 | $0.1<S\leq0.3$ |
| | | Big | 127 | $0.3<S$ |
| Sand boils | 4 | None | 275 | - |
| | | Less | 21 | - |
| | | Medium | 11 | - |
| | | Many | 135 | - |
| Ground crack | 2 | None | 106 | - |
| | | Yes | 336 | - |
| Lateral Spreading (m) | 4 | None | 437 | 0 |
| | | Small | 0 | $0<LS\leq0.1$ |
| | | Medium | 0 | $0.1<LS\leq0.3$ |
| | | Big | 5 | $0.3<LS$ |
| Severity of liquefaction-induced hazards | 4 | Little to None | 238 | - |
| | | Minor | 28 | - |
| | | Medium | 46 | - |
| | | Severe | 130 | - |



1    Table 3. Description of the severity of liquefaction-induced hazards.

| Severity of liquefaction-induced hazard | Description of field ground status |
|---|---|
| Little to None | Non-liquefaction. There is no sand boils phenomenon and no ground failure. |
| Minor | Slight liquefaction. The phenomenon of the sand boil is sporadic, but there is no ground failure. |
| Medium | Moderate liquefaction. There is a medium sand boil phenomenon, which has a short duration, small gushing quantity and small scale, the quantity of surface subsidence is less than 3% of the sand layer thickness that can cause structural damage, and the tiny crack of ground occurs, but there is no lateral spreading. |
| Severe | Serious liquefaction. There is a serious sand boil phenomenon, which has a long duration, large gushing quantity and wide scale, surface largely crazes, and lateral spreading and severe subsidence affect structures' services. The quantity of surface subsidence is more than 3% of the sand layer thickness. |




1    Table 4. Comparison of predictive performances of liquefaction-induced hazards.

| Category | Method | Accuracy | Brier score | Damage state | Recall | Precision | AUC of ROC |
|---|---|---|---|---|---|---|---|
| Ground cracks | BN | 0.909 | 0.070 | Yes | 0.742 | 0.920 | 0.780 |
| | | | | None | 0.975 | 0.920 | 0.962 |
| | ANN | 0.873 | 0.091 | Yes | 0.581 | 0.947 | 0.641 |
| | | | | None | 0.987 | 0.857 | 0.949 |
| Sand boils | BN | 0.918 | 0.106 | Many | 0.932 | 0.911 | 0.558 |
| | | | | Medium | - | - | - |
| | | | | Less | 0.857 | 0.857 | 0.667 |
| | | | | None | 0.932 | 0.948 | 0.982 |
| | ANN | 0.736 | 0.130 | Many | 0.591 | 0.813 | 0.652 |
| | | | | Medium | - | - | - |
| | | | | Less | 0.000 | 0.000 | 0.000 |
| | | | | None | 0.932 | 0.733 | 0.973 |
| Settlement | BN | 0.836 | 0.110 | Big | 0.867 | 0.703 | 0.845 |
| | | | | Medium | 0.815 | 0.957 | 0.745 |
| | | | | Small | 1.000 | 0.600 | 1.000 |
| | | | | None | 0.840 | 0.933 | 1.000 |
| | ANN | 0.745 | 0.130 | Big | 0.667 | 0.741 | 0.815 |
| | | | | Medium | 0.444 | 0.857 | 0.542 |
| | | | | Small | 0.000 | 0.000 | 0.000 |
| | | | | None | 1.000 | 0.735 | 1.000 |
| | I&Y Simplified Method | 0.727 | - | Big | 0.862 | 1.000 | - |
| | | | | Medium | 0.778 | 0.840 | - |
| | | | | Small | 0.667 | 0.069 | - |
| | | | | None | 0.600 | 1.000 | - |
| Lateral spreading | BN | 0.955 | 0.024 | Big | 1.000 | 0.286 | 1.000 |
| | | | | Medium | - | - | - |
| | | | | Small | - | - | - |
| | | | | None | 0.954 | 1.000 | 1.000 |
| | ANN | 0.982 | 0.018 | Big | 0.000 | - | 0.000 |
| | | | | Medium | - | - | - |
| | | | | Small | - | - | - |
| | | | | None | 1.000 | 0.982 | 1.000 |
| Severity of liquefaction-induced hazards | BN | 0.936 | 0.124 | Severe | 0.935 | 0.967 | 0.879 |
| | | | | Medium | 0.857 | 0.900 | 0.626 |
| | | | | Minor | 0.875 | 0.700 | 1.000 |
| | | | | None | 0.980 | 0.980 | 0.980 |
| | ANN | 0.718 | 0.117 | Severe | 0.710 | 0.710 | 0.785 |
| | | | | Medium | 0.333 | 0.636 | 0.776 |
| | | | | Minor | 0.000 | 0.000 | 0.000 |
| | | | | None | 1.000 | 0.746 | 1.000 |





1    Table 5. The posterior probabilities of partial output variables

| Output variable | Grade | Risk probability 1 | Risk probability 2 | Risk probability 3 | Risk probability 4 |
|---|---|---|---|---|---|
| Liquefaction potential | Yes | 0.572 | **1** | **1** | **1** |
| | None | 0.428 | **0** | **0** | **0** |
| Liquefaction potential index | Serious | 0.220 | 0.385 | **1** | **1** |
| | Moderate | 0.257 | 0.450 | **0** | **0** |
| | Slight | 0.207 | 0.136 | **0** | **0** |
| | None | 0.316 | 0.286 | **0** | **0** |
| Ground cracks | Yes | 0.239 | 0.408 | 0.661 | **1** |
| | None | 0.761 | 0.592 | 0.339 | **0** |
| Sand boils | Many | 0.304 | 0.515 | 0.696 | **1** |
| | Medium | 0.0554 | 0.0809 | 0.0676 | **0** |
| | Less | 0.0506 | 0.0725 | 0.0355 | **0** |
| | None | 0.590 | 0.331 | 0.201 | **0** |
| Lateral Spreading | Big | 0.076 | 0.0811 | 0.095 | 0.0927 |
| | Medium | 0.0698 | 0.0702 | 0.0764 | 0.0782 |
| | Small | 0.0698 | 0.0702 | 0.0764 | 0.0782 |
| | None | 0.784 | 0.778 | 0.748 | 0.751 |
| Settlement | Big | 0.255 | 0.362 | 0.498 | 0.523 |
| | Medium | 0.179 | 0.229 | 0.140 | 0.120 |
| | Small | 0.162 | 0.198 | 0.212 | 0.240 |
| | None | 0.404 | 0.212 | 0.150 | 0.116 |
| SLH | Severe | 0.277 | 0.416 | 0.641 | 0.746 |
| | Medium | 0.168 | 0.225 | 0.0966 | 0.697 |
| | Minor | 0.140 | 0.174 | 0.147 | 0.114 |
| | None | 0.415 | 0.185 | 0.116 | 0.697 |

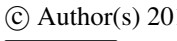



1 Table 6. The most probable explanation of LP, serious LPI, GC, many SB, big LS and

2 big S in the BN model

| Factor | LP | LPI | GC | SB | LS | S |
|---|---|---|---|---|---|---|
| Earthquake magnitude | Super | Super | Super | Super | Super | Super |
| Epicentral distance | Medium | Medium | Medium | Medium | Medium | Medium |
| Duration of earthquake | Medium | Medium | Medium | Medium | Medium | Medium |
| PGA | Medium | Medium | Medium | Medium | Higher | Higher |
| Fines content | Medium | Medium | Medium | Medium | Medium | Medium |
| Soil type | SM | SM | SM | SM | SP | SP |
| $D_{50}$ | Medium | Medium | Medium | Medium | Medium | Medium |
| SPT No. | Loose | Loose | Loose | Loose | Loose | Loose |
| $\sigma'$ | Small | Small | Small | Small | Small | Small |
| Groundwater table | Shallow | Shallow | Shallow | Shallow | Shallow | Shallow |
| Depth of soil layer | Shallow | Shallow | Shallow | Shallow | Shallow | Shallow |
| Thickness of soil layer | Thin | Medium | Medium | Thin | Medium | Thin |
| LP | - | Yes | Yes | Yes | Yes | Yes |
| LPI | - | - | Serious | Moderate | Serious | Moderate |
| GC | - | - | - | None | Yes | None |
| SB | - | - | - | - | Many | Many |
| LS | - | - | - | - | - | None |





1    Table 7. Sensitivity analysis of seismic liquefaction-induced hazards.

| Factor | Mutual information | | | | |
|---|---|---|---|---|---|
| | GC | SB | S | LS | SLH |
| The magnitude of earthquake | 0.002 | 0.003 | 0.002 | 0.001 | 0.002 |
| Epicentral distance | 0.004 | 0.007 | 0.007 | 0.002 | 0.008 |
| The duration of earthquake | **0.008** | **0.016** | **0.013** | 0.002 | **0.015** |
| PGA | 0.004 | **0.011** | **0.029** | **0.123** | **0.026** |
| Fines content | 0.001 | 0.001 | 0.001 | 0.003 | 0.001 |
| Soil type | 0.001 | 0.002 | 0.006 | **0.013** | 0.005 |
| $D_{50}$ | **0.009** | 0.001 | 0.002 | **0.029** | 0.003 |
| SPTN | 0.004 | **0.017** | **0.017** | 0.003 | **0.019** |
| $\sigma_v'$ | 0.001 | **0.010** | 0.007 | 0.006 | 0.008 |
| Groundwater table | 0.000 | **0.054** | 0.003 | 0.002 | 0.004 |
| Depth of soil deposit | **0.013** | **0.010** | **0.009** | **0.014** | **0.010** |
| The thickness of soil layer | **0.035** | **0.023** | 0.006 | **0.028** | 0.005 |





Natural Hazards
and Earth System
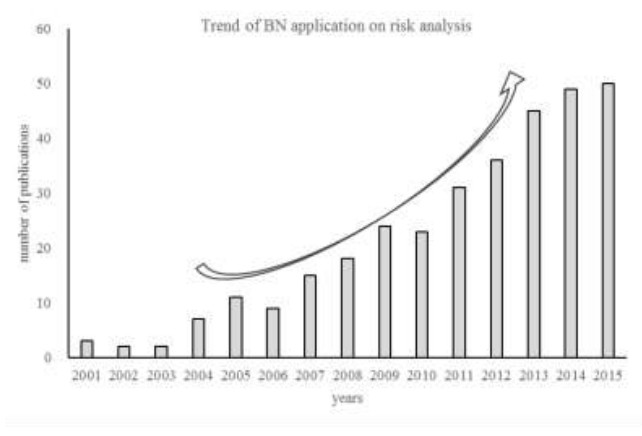

3 Figure 1. Increasing application of BN in risk analysis (update Weber et al. 2012).

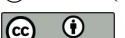



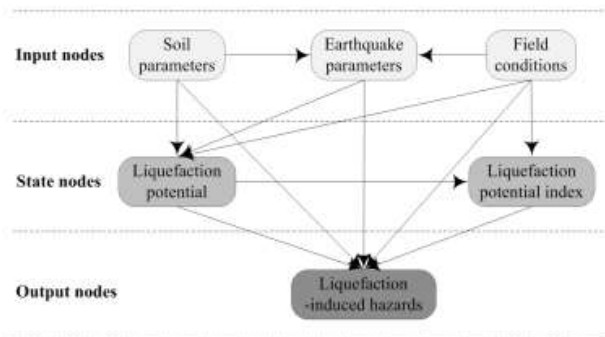

3     Figure 2. A generic BN for liquefaction-induced hazards.




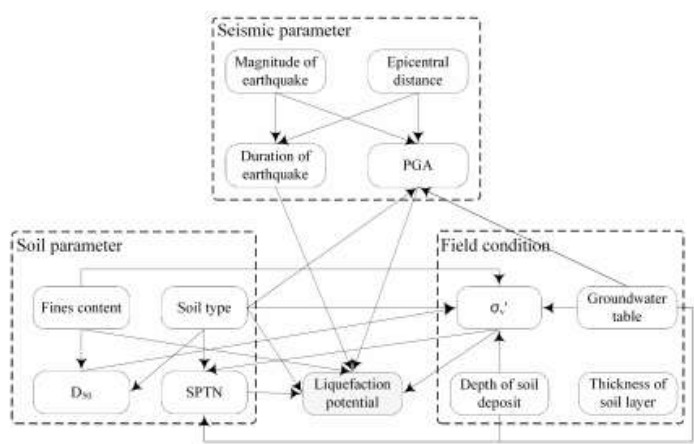

3    Figure 3. A BN model of seismic liquefaction (Hu et al. 2016).



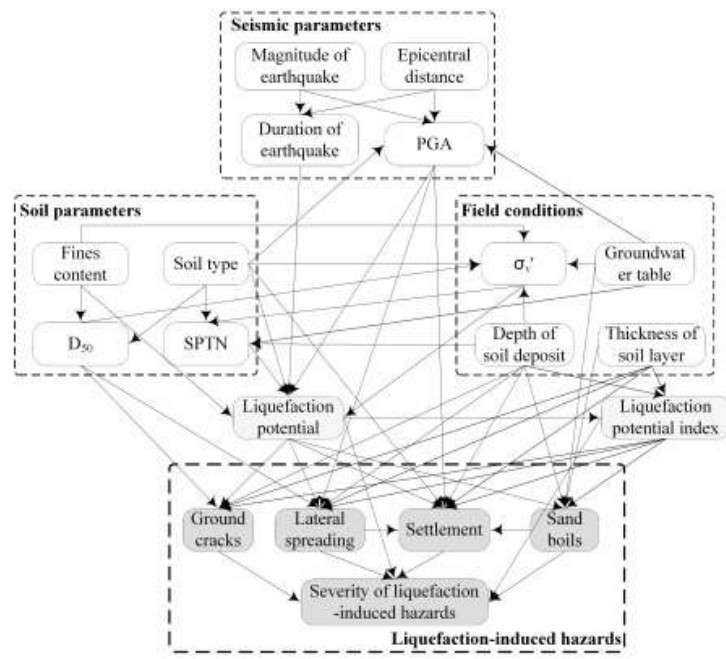

3    Figure 4. A BN model of seismic liquefaction-induced hazards.



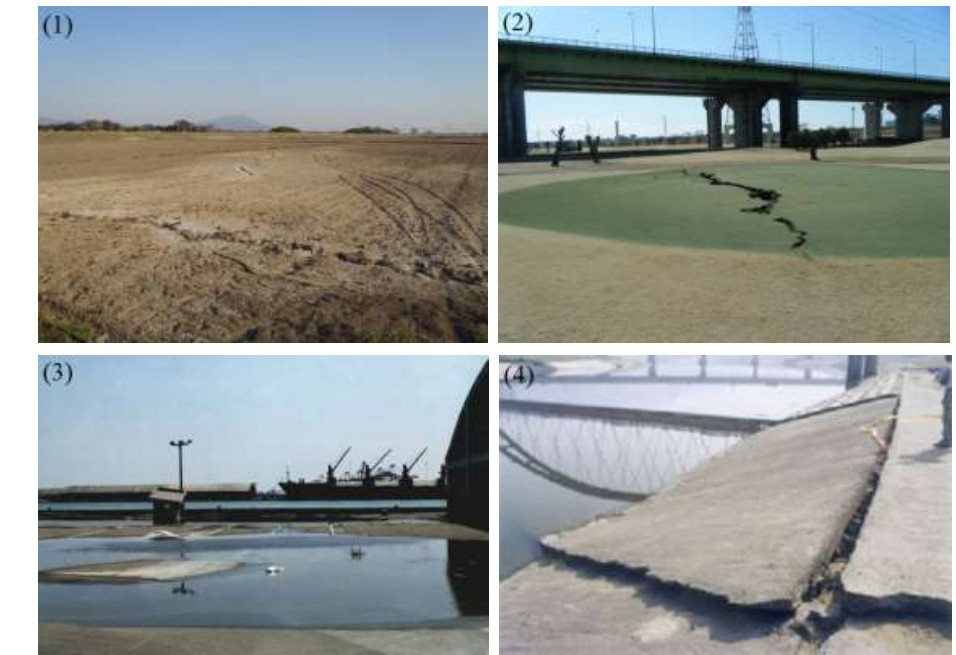

Figure 5. Photos showing liquefaction-induced hazards during the Chi-chi earthquake
and the 2011 Tohoku earthquake: (1) Sand boils in Chikusei city; (2) Ground cracks at
Arakawa    River    in    Toda    city;    (3)    Settlement    at    Taichung    Port
(http://www.ces.clemson.edu/chichi/TW-LIQ/Liq-Album/Settlement-7.htm); (4) Lateral
spread       induced       failure       of       a       dike       in       Nantou
(http://www.ces.clemson.edu/chichi/TW-LIQ/Liq-Album/LatSpread-3.htm).



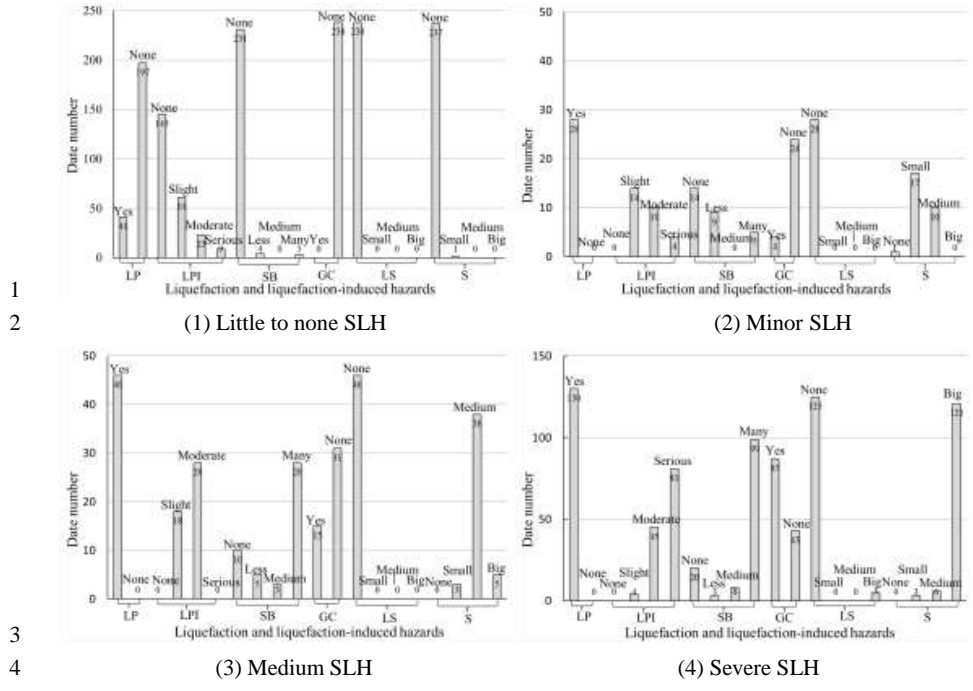

2            (1) Little to none SLH                 (2) Minor SLH

4              (3) Medium SLH                 (4) Severe SLH

6     Figure 6. Statistic summary of seismic liquefaction-induced hazards data.



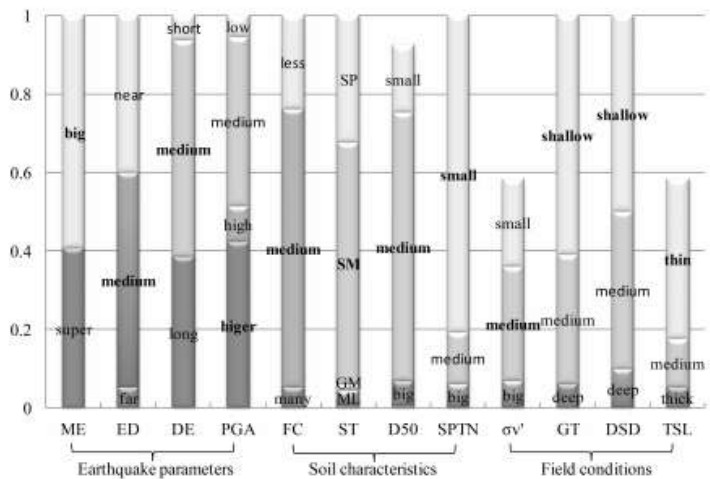

3  Figure 7. Ratios of all influence factors for the severe status of SLH



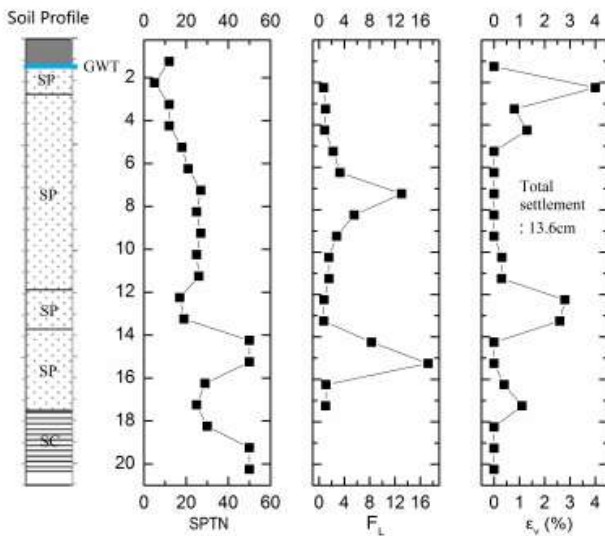

3    Figure 8. Soil profile and estimate of settlement




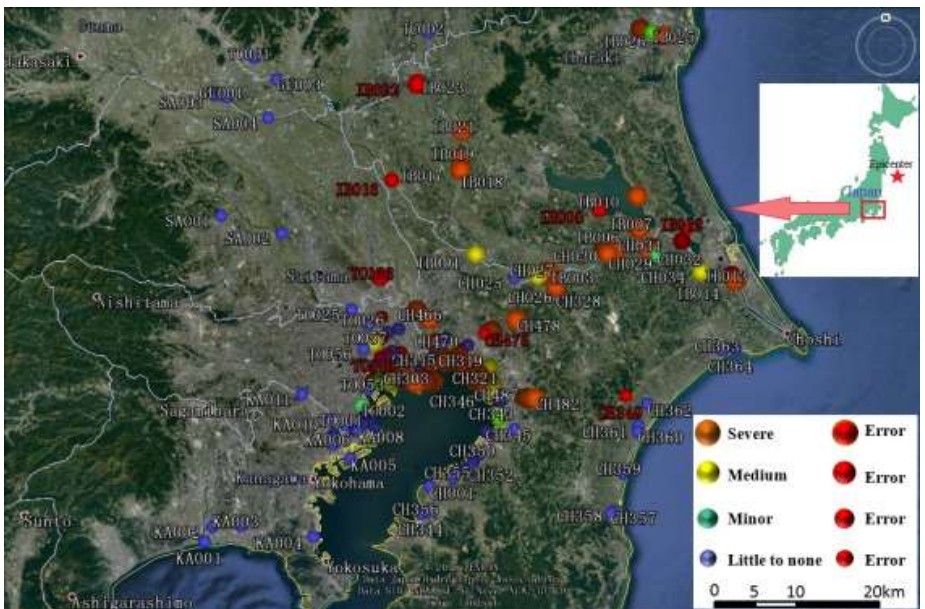

Figure 9. Assessment results of the severity of hazards induced by seismic liquefaction

in the northeast area of Japan in the 2011 Tohoku earthquake.