# Peer review of "Assessment of liquefaction-induced hazards using Bayesian networks"

_Natural Hazards and Earth System Sciences, 2017_

## Referee Comment (RC1) · R. E. S. Moss (Referee) · 10 Oct 2017

I'm interested in reviewing this manuscript but the English grammar and usage are not adequate enough to proceed. Please revise with a native English speaker and resubmit.
* * *

---

## Author Comment (AC1) · 11 Oct 2017

Dear Prof. Moss, Thanks for your interest in this manuscript. We will revise the English grammar and usage of the manuscript by asking a native English speaker for help according your suggestion.
* * *

---

## Referee Comment (RC2) · Anonymous Referee #2 · 20 Nov 2017

This paper deals with the development and the applicability of a Bayesian Network-based approach to assess soil liquefaction-induced hazards. The approach is interesting, and the subject fits into the scope of the journal. However, the current status of the submission does not permit a detailed review of this paper at this stage. This is mostly related to the fact that although the paper is readable, the language of the article is highly imprecise and generally confusing over large passages. The introductory section should be rearranged and shortened, also merging Section 2 with the introduction. This would avoid several repetitions in the text. Additionally, the data used in this study should be described in more detail, supported by suitable figures, statistics and graphs. It is hard for the reader to verify the results of the analysis as described in the

text since insufficient information on data properties and analysis results are provided. The Figures (especially Figure 9) are of inacceptable quality.

I would recommend the authors to substantially rework their paper in terms of the English language used, having a native speaker spell-checked the article before re-submission. Additionally, the authors should better describe the data they've used and the results of the analysis. When successfully resubmitted, the article can be reviewed in more detail.

---

## Author Comment (AC3) · 24 Nov 2017

Thanks for your interest in this manuscript. We have revised the English grammar and usage of the manuscript by asking a native English speaker for help according to your suggestion, the improved language certificate can be available in the supplement file 2. The details of revised parts are as follows:

(1) The introductory section has been rearranged and shortened according to your suggestion, the content is controlled in two pages. The part content of Section 2 was merged into the Section 3.

[Figure]

(2) The datum in this study was described in more detail by both texts and figures (see Fig. 5 in the supplement file 1) to provide more information about liquefied sites and severe damages to sites. The new content in Section 3.1 is as follow:

Only four earthquakes above are considered in this study, 'Strong magnitude' (6<Mw <7) is not included. The collected datum of these four earthquakes covers not only different duration and PGA, but also several soil parameters and field conditions, none of which is located within 10 km (defined as 'Near' epicentral distances) from earthquake sources. The grading standard of all 12 influence factors of liquefaction potential in Fig. 5 can reference Hu et al. (2016). The observed liquefaction effects induced by these earthquakes include sand boils, settlement of ground, ground cracks, and lateral spreading, resulting in the destruction of cropland, blocking of channels, and severe damage or collapse of many buildings, highways, bridges, harbour facilities, and other infrastructure components. The liquefied sites of the collected datum in this study are mainly contained in Chi-Chi earthquake and Tohoku earthquake. The characteristics of liquefied soils are predominantly loose and clean sands or silty sands (SPT values less than 10) that deposit within 10 meters in the liquefaction datum of the two earthquakes shown in Fig. 5(2). It is worth noting that duration of ground motion was very long within 100-200s, and the liquefied sites were very far from the epicentre of about 300-450km experienced peak ground accelerations of approximately 150-300cm/s2 in Tohoku earthquake, whereas serious damage induced by soil liquefaction occurred in a wide area of the Tohoku and the Kanto regions along with wide scale of sand boils, cracks and severe uneven settlement of pavements due to cycle shear actions for a long time. However, in Chi-Chi earthquake, durations of the strong motions were short, but PGA values were very big due to near a source earthquake proximal to a fault (proximately 1.0km), e.g. in the Nantou and Wufeng regions as high as 0.7-1.0g, that caused widespread liquefaction in the forms of sand boils, lateral spreads, and settlement of grounds in the towns of Yuanlin, Nantou, and Wufeng, Taiwan. Fig. 5(3) shows proportions of all influence factors for the severe status of the SLH. It is easily seen that most severe damage sites suffered from big or super earthquakes (Mw > 7

or 8) with long loading (duration more than 60s), some epicentral distances were close to the earthquake sources, e.g. the nearest liquefied sites in Nantou city are about 14km away from the epicentre, thus their PGA was sufficiently high. As for soil characteristics, pure sand or silty sand with moderate fine content (30% $<$FC$\leq$ 50%) and moderate average grain diameter (0.075$\leq$D50$<$0.425) values result in severe damage, unlike sites with gravelly soil and sandy silt. The damage phenomena also indicate that, even though gravel and sandy silt are not easily liquefied if the earthquake is sufficiently strong to cause liquefaction, severe damage can be expected shown in Fig. 5(2) and (3). The small SPT number (0 $<$ SPTN $\leq$10) means that the sandy soil is so loose that settlement and lateral spreading are more likely triggered after liquefaction because loose sand is easier to be compressed and flow during seismic liquefaction. As for field conditions, the shallow-buried sandy soil layer has low effective stress ($\sigma$v' $<$ 50kpa) and the groundwater table is near to the ground surface. Such zones are likely to suffer from severe damage. The above laws fit well with practical engineering experience. The sum of the data size of these twelve variables is not consistent in Fig. 5(1), (2), and (3) respectively, such as epicentral distance, duration of the earthquake, D50, $\sigma$v', and the thickness of the soil layer due to the missing data. The proportion of missing data for epicentral distance, duration of the earthquake, D50, vertical effective stress, and the thickness of soil layer are $\sim$5%, $\sim$9.7%, $\sim$15.2%, $\sim$29.4%, and $\sim$38.9%, respectively.

(3) Qualities of all figures in this study have been improved with 300dpi that can be found in the supplement file 1.

Please also note the supplement to this comment:
https://www.nat-hazards-earth-syst-sci-discuss.net/nhess-2017-80/nhess-2017-80-AC3-supplement.zip

---

## Author Response (AR1)

Thanks a lot for the two reviewers' comments. We have revised the English grammar and usage of the manuscript by asking a native English speaker for help according to their suggestions. The improved language certificate can be available in the supplement file. The details of revised parts are as follows:

(1) The introductory section has been rearranged and shortened according to the suggestion form the referee #1, the content is controlled in two pages. The part content of Section 2 was merged into the Section 3.

(2) The data in this study was described in more detail by both texts and figures (see Fig. 5 in the revised manuscript) to provide more information about liquefied sites and severe damages to sites. The added context can be found in Section 3.1 that was marked in red.

(3) Qualities of all figures in this study have been improved with 300dpi that can be found at end of the manuscript.

[Figure]

**Language Editing Services**

*Registered Office:*
Elsevier Ltd
The Boulevard, Langford Lane,
Kidlington, OX5 1GB, UK.
Registration No. 331566771

**To whom it may concern**

The paper "Risk assessment of liquefaction-induced hazards using Bayesian network based on standard penetration test data" by Tang Xiaowei, Bai Xu, Qiu Jiangnan, Hu Jilei was edited by Elsevier Language Editing Services.

Kind regards,

**Elsevier Webshop Support**

(This is a computer generated advice and does not require any signature)

---

## Author Response (AR2)

Dear Editor and Reviewers:

I quite appreciate your favorable consideration and the reviewer's insightful comments concerning our manuscript entitled "Assessment of liquefaction-induced hazards using Bayesian networks based on standard penetration test data" (No. nhess-2017-80). Those comments are all valuable and very helpful for revising and improving our manuscript, as well as the important guiding significance to our research. Now, we have made correction exactly according to the reviewer's comments, which we hope to meet with approval.

Once again, thank you very much for your comments and suggestions. The main corrections in the paper and response to the reviewer's comments are as follows:

Yours sincerely,

Xiaowei TANG , Xu BAI, Jilei HU, and Jiangnan QIU

Report #1:

**Suggestion for revision:**

1. Page 2, line 12, Liquefaction empirical models, please try to include research by (Goh 1994; Zhang and Goh 2013, Pal 2006; 13 Toprak et al. 1999, Zhang et al. 2015, Zhang and Goh 2016) based on historical data: Zhang WG, Goh ATC. 2013. Multivariate adaptive regression splines for analysis of geotechnical engineering systems. Computers and Geotechnics 48: 82-95. Zhang WG, Goh ATC, Zhang YM, Chen YM, Xiao Y. 2015. Assessment of soil liquefaction based on capacity energy concept and multivariate adaptive regression splines. Engineering Geology 188: 29-37. Zhang WG, Goh ATC. 2016. Multivariate adaptive regression splines and neural network models for prediction of pile drivability. Geoscience Frontiers. 7: 45-52.

**Response:** It is very grateful to your suggestion. These proposed papers by the reviewer are very valuable and helpful to our manuscript. The authors had studied them carefully and added them to Line 12-13, Page 2 in Introduction and Line 20-25, Page 23 in References.

2. Page 11, Furthermore, there are no effective simplified methods for estimating ground cracks and sand boils, and simplified methods for calculating lateral spreading

(Bartlett and Youd 1995, Wang and Rahman 1999, Goh et al. 2014) require the free face ratio or ground slope, which were not included in the data collected for this study. Please check Goh et al. 2014, the reviewer think it is Goh and Zhang 2014.

**Response**:Thanks for your careful inspection. The authors indeed made a mistake of the author's ranking of the reference (Goh and Zhang 2014). The authors revised it as shown on Line 19, Page 11.

3. About the use of liquefaction potential index LPI, other researchers have used the similar term-seismic liquefaction potential, the authors are encouraged to include MARS_LR approach: Zhang WG, Goh ATC*. 2016. Evaluating seismic liquefaction potential using multivariate adaptive regression splines and logistic regression. Geomechanics and Engineering, 10(3): 269-284.

**Response**:The authors studied the proposed paper and found that Zhang and Goh (2016) solved the liquefaction classification problem using the Multivariate Adaptive Regression Splines (MARS) approach based on Logistic Regression, rather than a calculation of liquefaction potential index (Iwasaki et al. 1982). In addition, the liquefaction potential index (LPI) was taken as an input variable for assessing liquefaction-induced hazards.

4. Line 21, Page 15, discussion part, it should be noted that the Goh and Zhang (2014) method for estimating lateral spreading is the multivariate adaptive regression splines (MARS) method instead of the multiple linear regression (MLR).

**Response**:The authors revised the mistake according to your suggestion as shown in Line 21, Page 15. Thanks a lot for your kind suggestion again.

5. As is well known, the ANN is a black-box method, that is, quite difficult to interpret the built model (maybe mainly through the weights, bias values, as well as the transfer functions). The authors should also explain the interpretability of the developed BN model and compare it with the MARS method and the empirical equations.

**Response**:In the Discussion,the authors explain the interpretability of the developed BN model and compare it with the MARS method from Line 24-26, Page 16 to Line 1-4, Page 17: Both the BN model and the MARS model are probability models which can possess interpretability in mathematics,unlike the ANN method with "black-box"

technology. They can easily develop comprehensive models that take into consideration all the independent variables with highly nonlinear. However, The MARS model reflects the functional relationship between the output parameter and the independent variables, and its equation form should be known at first before constructing the model. Additionally, the MARS model can only predict a single output (e.g. liquefaction potential or lateral spreading) at one time, whereas the BN model can reflect causalities or logical relationships among all the variables in graphically without any mathematical expression, and it also can predict several outputs (e.g. liquefaction potential, settlement, and lateral spreading, etc.) simultaneously and can proceed construction model and prediction under condition of missing values using the EM algorithm. It is worth to note that the main difference between the two models is that the BN model can skillfully combine with the prior knowledge and evidence (e.g. liquefaction data) by Bayes' formula that can improve the prediction accuracy of the BN model, but the prediction of the MARS model only depends on collected data.

6. As a data-driven method, what is the main limitation of the proposed BN model?

**Response**:The limitations of the BN method are that it needs a mass of data when constructing a BN model to guarantee a certain accuracy, if relative small amount of data are collected, it easily results in a non-robust BN model structure; and the causality or the logical relationship between two variables in a BN model obtained only by the data-driven algorithm is sometimes acceptable in mathematics, but not true in physics. The authors added the content from Line 5 to Line 9, Page 17.

Report #2:

**Suggestion for revision**:

I would only like to advise the authors to remove the word "risk" from the title since this is misleading as the study does not deal with risk assessment (e.g., "Assessment of liquefaction-induced hazards using Bayesian networks based on standard penetration test data")

**Response**:The authors quite appreciate your suggestion for the revision of the misleading statement in the title. The revised statement of the title is shown in Line 1, Page 1.

**References**

Iwasaki T., Tokida K., Tatsuoka F., et al. Microzonation for soil liquefaction potential using simplified methods. Proc. 3rd International Earthquake Microzonation Conference, Seattle, 1319-1330, 1982.

---

## Author Response (AR3)

Dear Editor:

I quite appreciate your patience and detail oriented on our manuscript entitled "Assessment of liquefaction-induced hazards using Bayesian networks based on standard penetration test data" (No. nhess-2017-80). Those comments about the figures are all valuable and very helpful for revising and improving our manuscript, as well as the important guiding significance to our research. Now, we have made correction exactly according to the editor's comments, which we hope to meet with approval.

Once again, thank you very much for your comments and suggestions. The main corrections in the paper and response to your comments are as follows:

Yours sincerely,

Xiaowei TANG , Xu BAI, Jilei HU, and Jiangnan QIU

**Suggestion for revision:**

1. In Figure 5, the dataset used for the case study is shown. However, as the meaning of the x-axe labels can be derived from the text (not from the caption), the individual bar labels can hardly be read and their meanings are sometimes not clear (e.g., GT: what is "deep", "medium", "shallow"?) Please accompany the Figure with an appropriate table or similar to let the reader now about the parameter class ranges.

**Response:** It is very grateful to your suggestion. In previous versions of our manuscript, the authors did not show the specific instruction of the grading standards for the 12 factors. It is not visualized enough to understand them just according to the reference paper behind. We had added a new Table 2 to introduce them in Page 25-26 and corrected the statement in Line 29, Page 7 in the revision. The serial number of tables behind is also changed from 2-7 to 3-8 simultaneously.

2. In addition, Figure 9 should be reworked as the point labels are not readable, sometimes masking the site info. Please check if the labels are really needed, and modify the Figure in such that what is written in the text (Section 5) can easily be verified by the reader.

**Response:** Figure 9 helps to visualize the assessment results of the severity of hazards. It does not preclude the understanding of Section 5 without figure 9. We delete this figure and the statement about it in Section 5 according to the editor's comment.